# The Regulation of Nodule Number in Legumes Is a Balance of Three Signal Transduction Pathways

**DOI:** 10.3390/ijms22031117

**Published:** 2021-01-23

**Authors:** Diptee Chaulagain, Julia Frugoli

**Affiliations:** Department of Genetics & Biochemistry, Clemson University, Clemson, SC 29634, USA; dchaula@g.clemson.edu

**Keywords:** autoregulation of nodulation, nodulation, nitrogen response in nodulation, *Medicago truncatula*

## Abstract

Nitrogen is a major determinant of plant growth and productivity and the ability of legumes to form a symbiotic relationship with nitrogen-fixing rhizobia bacteria allows legumes to exploit nitrogen-poor niches in the biosphere. But hosting nitrogen-fixing bacteria comes with a metabolic cost, and the process requires regulation. The symbiosis is regulated through three signal transduction pathways: in response to available nitrogen, at the initiation of contact between the organisms, and during the development of the nodules that will host the rhizobia. Here we provide an overview of our knowledge of how the three signaling pathways operate in space and time, and what we know about the cross-talk between symbiotic signaling for nodule initiation and organogenesis, nitrate dependent signaling, and autoregulation of nodulation. Identification of common components and points of intersection suggest directions for research on the fine-tuning of the plant’s response to rhizobia.

## 1. Introduction

Nitrogen (N) is a major determinant of plant growth and productivity. In addition, N is required as a constituent of nitric oxide (NO) and polyamines that influence constitutive and induced plant defense [1]. While N is the most abundant gas in the atmosphere, it is unusable as a direct source of plant nutrients because of the inability of plants and most organisms to enzymatically break the triple bond of N_2_ and convert it into the main forms that plant roots can take up: NO_3_^-^ and NH_4_^+^. Thus, N as a plant nutrient must be obtained from decomposition products in the soil or added to soil in plant-absorbable forms. 

The largest natural source of N input to the biosphere is biological nitrogen fixation, adding approximately 50–70 Tg of N globally to agricultural systems [2]. Biological nitrogen fixation is the conversion of N_2_ to NH_3_ catalyzed by nitrogenase enzyme in diazotrophs. These diazotrophs are both free-living and in symbiotic associations between plants and nitrogen-fixing bacteria (legume-rhizobia, *Azolla*-cyanobacteria, nonlegume-*Frankia*). A smaller amount of N input to the biosphere is contributed by nitrates in the rainwater and by organic nitrogen through manure. Non-legume plants take up on average 20–50 g of N per1 Kg of dry biomass produced [3]. In contrast, soybean, a widely cultivated legume for human consumption and animal feed, grown in unfertilized soil contains 55–70% of fixed nitrogen in its aboveground parts during the nodulation period [4]. Thus, symbiotic nitrogen fixation (SNF) is of intense interest as an alternative to chemical fertilizer. Because only a small proportion of commercial legume crop production relies on biological nitrogen fixation, a better understanding of the legume-rhizobia symbiosis could enable the efficient use of the natural process of SNF and reduce dependence on chemical N fertilizer. 

SNF is the result of a mutualistic interaction between a compatible plant and diazotrophs in which the plant provides a niche and fixed carbon to bacteria in exchange for fixed nitrogen. The plant family Fabaceae (Legumes) is the third-largest family of flowering plants consisting of ~19,000 known species, 88% of which form nitrogen-fixing root nodules in symbiosis with rhizobia [5]. Legumes are commonly cultivated as food crops, forage, or green manure; *Glycine max* (soybean), *Phaseolus vulgaris* (bean), *Arachis hypogaea* (peanut), *Medicago sativa* (alfalfa) are just a few among many cultivated legumes. Combined with two legumes adopted as model systems, *Medicago truncatula* and *Lotus japonicus*, genetic studies have led to a wealth of information on the signaling involved in establishing and regulating nodule development [6]. This review addresses two systemic pathways, autoregulation of nodulation (AON), which involve control of nodule numbers in response to the establishment of symbiosis, the systemic pathway that controls nodule number in response to available soil nitrogen and addresses what is known about how the two systemic pathways interact with the local pathway for the initiation of nodules. The cross-talk between initiation of nodulation pathways and the inhibitory pathways to prevent excess nodulation occurring at the same time results in a complex array and intertwining of signals that are just beginning to be understood and appreciated.

## 2. Signaling to Initiate and Form Symbiotic Nodules

Legumes like *M. truncatula*, *M. sativa*, and *Pisum sativum* form indeterminate nodules (contain persistent meristem resulting in cylindrical shaped nodules), whereas legumes like *Lotus japonicus*, *Glycine max*, and *Phaseolus vulgaris* form determinate nodules (spherical nodules lacking persistent meristem) [7]. The indeterminate nodules formed by *M. truncatula* are initiated from the inner cortex next to xylem poles, and at maturity, the meristem continues to produce new cells that are eventually infected, resulting in five developmental zones within a nodule. Named as follows, from the distal to proximal end of a nodule, they are (i) the meristematic zone followed by (ii) the invasion zone-characterized by actively growing infection threads, (iii) the interzone-consisting of differentiating bacteroids, (iv) the N-fixing zone-the site of mature bacteroids in the symbiosomes that fix nitrogen, and (v) the senescence zone- containing old degrading symbiosome [7]. In contrast, determinate nodules are initiated from cell division in the outer cortex; at maturity, nodules contain a relatively homozygous bacteroid population for nitrogen fixation and the nodules senesce within a few weeks of initiation [7]. Both nodule types contain leghemoglobin, providing pink coloration to nodules, which creates a near-anoxic environment in the nodule for nitrogen fixation by the oxygen intolerant nitrogenase enzyme [8]. Despite the differences in nodule type, *M. truncatula* and *L. japonicus* share many known genetic components involved in nodule formation and autoregulation of nodulation and are used as models for their respective nodulation types. The following summary focuses on *M. truncatula* (Mt) nodulation for simplicity, referring to *L. japonicus* (Lj) or other legumes *G. max* (Gm) and *P. vulagaris* (Pv) only where knowledge from the system has been applied to bridge the gap in knowledge in *M. truncatula* or the well-documented difference in between the two systems is considered important for this review. 

### 2.1. Symbiotic Partner Selection

The legume-rhizobia symbiosis for both models relies on chemical communication between the plant and the microbe. The communication starts as plant roots constitutively secrete specific flavonoids, which act as signals to rhizobia, inducing the production of rhizobial nodulation factors (Nod factors) [9,10] (Figure 1). Nod factors are lipochitooligosaccharides decorated by different substituents like methyl, fucosyl, acetyl, etc., giving them unique chemical structures [11,12]. The recognition of specific Nod factors by specific receptors in plants is the major determinant of host-rhizobia specificity [11].

### 2.2. Early Symbiotic Signaling

Nod factors (NFs) are perceived by LysM receptor-like kinases (LYKs) in root epidermal cells. Two forms of LysM receptor-like kinases play a role: MtLYK3/MtLYK4 [13] and MtNFP (NOD FACTOR PERCEPTION) [14,15]. Downstream of NF perception, three members of the MtCNGC15 family (CYCLIC NUCLEOTIDE GATED CHANNEL): CNGC15a, CNGC15b, and CNGC15c, form a complex with the potassium gated ion channel MtDMI1 (DOES NOT MAKE INFECTION 1) in the nuclear envelope to modulate nuclear calcium release [16]. The resulting calcium oscillation initiates a nuclear signaling cascade. The first step is the activation of the nuclear-localized calcium and calmodulin-dependent serine/threonine kinase MtDMI3 [17,18]. The Ca^2+^ influx and Cl^-^ and K^+^ efflux also results in cytoskeletal changes leading to root hair curling and entrapping of bacteria to form nodulation foci (Figure 1) (see Roy et al. [6] for review of genes involved in cytoskeletal changes described below). 

For successful nitrogen-fixing nodules, rhizobia must reach the newly divided cortical cells that eventually develop into a nodule (discussed in nodule organogenesis section below). Rhizobia trapped in the root hair curl enter the cell by the degradation of the cell wall and move through the root hair epidermal cell into the cortex by invagination of plasma membrane forming infection thread. The formation of an infection thread requires the expression of early nodulin genes and is facilitated by transcription factor signaling in the epidermal root hair cell nucleus (Figure 1). The MtDMI3 protein, activated by calcium spiking, binds to the transcription factor MtIPD3 (INTERACTING PROTEIN OF DMI3) [19] (CCaMK-CYCLOPS in *Lotus japonicus*) and activates it by phosphorylation [20]. A mutant of *MtDMI3*, *dmi3**-1*, results in plants that fail to produce infection threads and cortical cell division, leading to an absence of nodulation phenotype [21]. In contrast, *ipd3-1* mutant plants have delayed, defective infection threads, and nodulation is delayed, eventually forming non-infected nodules [22]. Interestingly, *MtIPD3L* (*IPD3-LIKE*) gene expression under the *MtIPD3* promoter rescued the *ipd3-2* mutant phenotype, showing a functional redundancy between the genes, but *ipd3l* mutants make normal infection threads and wild type nodules, while the *ipd3l ipd3-2* double mutant phenocopies *dmi3-1* [23]. 

The two GRAS family transcription factors MtNSP1 and MtNSP2 (*NODULATION SIGNALLING PATHWAY*) are also required for nodulation [24,25] and work as homodimers [26]. Null mutants of *MtNSP1* or *MtNSP2* are non-nodulating, whereas a weak allele of *MtNSP2* harboring a non-conservative single amino acid change results in plants with small white nodules [24,25]. MtDELLA, a negative regulator of gibberellic acid signaling, controls rhizobial infection and nodule number potentially by linking the MtDMI3-MtIPD3 and MtNSP2-MtNSP1 complexes, forming a bridge between MtIPD3 and MtNSP2 [27]. Mutants of *MtDELLA1*, *MtDELLA2*, and *MtDELLA3* have reduced nodule numbers and nodule density compared to wild type [28]. Cytokinin also regulates rhizobial infection and nodule primordia formation in multiple ways. The transcription factor MtKNOX3 (KNOTTED-1 LIKE HOMEOBOX3) is upregulated during nodule initiation, and knockdown of *MtKNOX3* causes downregulation of type A cytokinin response genes [29]. Since MtKNOX3 binds to the promoters of the *LONELY GUY* genes *MtLOG1* and *MtLOG2*, as well as *MtIPT3* (*ISOPENTYL TRANSFERASE* 3), *MtKNOX3* may increase cytokinin synthesis in developing nodules [30]. The type-B Response Regulator (MtRRB3), another transcription factor involved in cytokinin signaling, interacts with and trans-activates MtNSP2 and Cell Cycle Switch 52A (MtCCS52A), supported by the observation that *rrb3* mutants form a lower number of infection threads and nodules [31]. The symbiotic pathway can be independently activated by MtDMI3-MtIPD3, MtNSP1-MtNSP2 and MtDMI3-MtIPD3-MtDELLA-MtNSP1-MtNSP2 complexes by upregulating expression of the transcription factor MtNIN (NODULE INCEPTION) [26,27,32]. 

Genetically positioned downstream of *NIN*, other transcription factors also act in the pathway, including *Nuclear Factor Y* consisting of a heterotrimeric complex of NF-YA, NF-YB, and NF-YC, and *ERN1* (*ETHYLENE RESPONSE FACTOR REQUIRED FOR NODULATION 1*) and *ERN2* (Figure 1 and reviewed by Roy et al. [6]). MtNIN and MtERN1 control the expression of cell wall-associated *MtENOD11 (EARLY NODULIN 11)* and *MtENOD12*, which are critical for infection thread development [33]. Furthermore, *MtENOD11* transcription is abolished in the *ern1* mutant [34], and upregulation of rhizobia-dependent *MtENOD11* is absent in *nfp, dmi1, dmi2, dmi3, nsp1*, and *nsp2* mutants [35], establishing early nodulin gene expression as crucial in symbiosis. Interestingly, as shown in Figure 1, *MtNIN* expression (dependent on cytokinin signaling through the MtCRE1 receptor) is sufficient to induce expression of two small peptides in the root epidermis: MtCEP7-involved in nodulation and MtCLE13-involved in inhibition of nodulation soon after rhizobial inoculation (detected at 4 h post-inoculation) and then later in the nodule primordia (4 days post-inoculation) [36]. 

Recently, chromatin remodeling has been preliminarily shown to be critical to the development of nodule primordia. In a report on bioRxiv, the Bisseling lab suggests *M. truncatula* histone deacetylases (MtHDTs) are required in nodule primordia, based on conditional RNAi [37]. However, this requirement appears to be because of the effect of reduced MtHDTs on a single gene: MtHDTs positively regulate 3-hydroxy-3-methylglutaryl coenzyme a reductase 1 (*MtHMGR1*) in a cell-autonomous manner [37]. The *MtHMGR* genes encode enzymes that catalyze the rate-limiting step in a pathway that synthesizes precursors to multiple plant hormones, including cytokinin, brassinosteriods, gibberellin, and abscisic acid [38] and *M. truncatula* plants carrying mutations in *HMGR1* do not initiate calcium spiking or form nodules [39].

### 2.3. Nodule Initiation and Organogenesis

Nodule organogenesis includes both bacterial colonization to form symbiosomes and the autoregulation of nodulation to control nodule number, which occurs simultaneously with the cell division leading to the development of the nodule as a visible root organ [6]. A gain of function mutation in the *L. japonicus HISTIDINE KINASE* gene (*LjLHK1*, a cytokinin receptor corresponding to *M. truncatula MtCRE1*) results in spontaneous nodule formation even in the absence of rhizobia, whereas loss of function mutation in the same gene causes hyperinfection and failure to timely initiate nodule primordia after rhizobia inoculation, indicating cytokinin signaling through LjLHK1 is sufficient for cell division leading to nodule development [40,41,42]. MtCRE1, the *M. truncatula* equivalent of LjLHK1, functions in the initial cortical cell division and later in the transition between meristematic and differentiation zones of the mature nodule [43]. MtCRE1 signaling also activates the downstream nodulation-related transcription factors MtERN1, MtNSP2, and MtNIN, as well as regulates the expression and accumulation of PINFORMED (MtPIN) auxin efflux carriers [40,43]. Furthermore, MtNIN is also required for the cortical cell division and progression of infection threads in cortical cells, as indicated by excessive nodulation foci and infection threads, but the absence of cell division and nodule primordia in a *MtNIN* promoter mutant lacking cis-regulatory cytokinin responsive elements [44]. Downstream of *MtNIN*, similar signaling events as described above in early symbiotic signaling lead to the activation of *ENOD11* (*EARLY NODULIN 11*) (Figure 1). In addition, the MtNSP1-MtNSP2 complex can bind directly to the *ENOD11* promoter to enhance its expression [25]. Both MtERN1 and MtERN2 function in the epidermis for infection thread development, whereas only MtERN1 is proposed to function in the cortex for nodule organogenesis. The *ern1* mutant displays limited root hair infection and cortical cell division, leading to growth-arrested non-infected nodules [34], whereas *ern2* (a mutant of *MtERN2*, a close sequence homolog of *MtERN1*) forms prematurely senescing nitrogen-fixing nodules that are partially defective in rhizobial colonization [45]. In contrast, *ern1 ern2* double mutant plants are impaired in root hair infection and do not display any symbiotic interactions, leading to the proposal that only MtERN1 functions in nodule organogenesis [45]. In addition to the role of transcription factors, nodule primordia formation in *M. truncatula* requires a local accumulation of auxin at the site of nodule initiation in the inner cortex, generated by inhibition of polar auxin transport (PAT) [46,47]. N signaling was recently linked to root growth in Arabidopsis through phosphorylation/dephosphorylation of PINs (auxin efflux carriers) [48], and this provides a way for PAT to be involved in both initiation and inhibition of nodule formation.

## 3. Signaling to Inhibit Nodule Formation

While the legume-rhizobia symbiosis involves the plant providing carbon and a niche for bacteria in exchange for nitrogen, the respiratory cost of a nodule associated with nitrogenase activity is 2–3 g carbon per nodule, whereas the total respiratory cost of nitrogen-fixing nodule is 3–5 g of carbon per nodule (or 6–12 g carbon per gram nitrogen) [49]. Increasing the number of nodules does not necessarily increase the total amount of nitrogen fixed, as observed in mutants that make a higher number of nodules (hypernodulating mutants) compared to wild type [50]. Thus, it is ecologically advantageous for legumes to suppress nodulation and SNF in the presence of soil nitrogen, as well as regulate the number of nodules formed according to the plant’s nitrogen requirements when growing in soil depleted of nitrogen. Two independent mechanisms exist for regulating nodulation: (i) N dependent control of nodulation, and (ii) autoregulation of nodulation (AON).

### 3.1. Control of Nodulation Based on Nitrogen Need and Soil Availability

High levels of soil N suppress nodulation and inhibit nitrogen fixation in already formed nodules rapidly after the addition of nitrogen fertilizer [51]. The response to soil nitrate in plants is both local and systemic. Nitrate transporters have an important role in nitrate sensing and nitrogen demand signaling in Arabidopsis (reviewed in [52]) but only two members of the NITRATE TRANSPORTER 1 (NRT1)/PEPTIDE TRANSPORTER (PTR) family have been studied in *M. truncatula*. Of the two, MtNPF6.8 affects nitrate-dependent regulation of primary root growth via abscisic acid signaling, suggesting its role as a nitrate sensor, whereas MtNPF1.7 is essential for nodule formation but does not have a known role in N demand signaling [53]. Externally sourced high N leads to higher shoot concentrations of N, resulting in increased shoot-to-root auxin transport in wild type, which is correlated with the reduced lateral root density. In contrast, nodule density in response to external N is not correlated to root-to-shoot auxin transport [54]. Two recent studies showed that expression of a small peptide of the CLAVATA3 (CLV)/EMBRYO SURROUNDING REGION (ESR)-RELATED (CLE) family in roots, MtCLE35, is induced in the presence of high nitrate and by rhizobia (Figure 2a) [55,56]. The overexpression of *MtCLE35* in roots reduces the nodule number in wild type systemically, depending on the MtSUNN receptor [55,56,57] demonstrated by the reduction of nodule number in non-transgenic roots of composite plants harboring transgenic roots overexpressing *MtCLE35* [55]. The *MtCLE35* overexpression was unable to reduce nodule number in *rdn1* mutant [56] suggesting MtCLE35 requires a similar post-transcriptional modification as MtCLE12 [56,58]. 

High N reduces the accumulation of miR2111 in both the shoot and root of the plant, and while *MtCLE35* overexpression reduces the accumulation of miR2111 in both the shoot and root, this accumulation is independent of external N availability, indicating nitrate control of nodulation shares the miR2111-TML components of the AON pathway of repression discussed in the next section [57]. Interestingly, the downregulation of *MtCLE35* using RNAi resulted in significant accumulation of miR2111 in the root but repression of only the *MtTML2* (TOO MUCH LOVE 2) transcript was observed [57]. Since miR2111 can target both *MtTML1* and *MtTML2* [59], the repression of only *MtTML2* combined with a partial but not complete bypass of N inhibition of nodulation by ectopic expression of miR2111, suggests the possibility of an alternate pathway for N inhibition or miR2111 independent differential post-transcriptional regulation of the two TMLs [57]. Thus, MtCLE35 is the systemic nitrate signal inhibiting nodulation in response to nitrate (Figure 2a) and also controls nodule number in AON in the presence of rhizobia (see AON section below and Figure 2c). Locally, in response to high nitrate, the NIN-LIKE PROTEIN (MtNLP1) re-localizes from the cytosol to the nucleus in root cells to inhibit rhizobial infection and nodule formation by physically interacting with MtNIN to suppress *CYTOKININ RESPONSE1* (*MtCRE1)* expression, thus inhibiting nodulation [60] (Figure 2a). Additionally, functional *MtNLP1* is required for N-dependent induction of the *MtCLE35* transcript [57], suggesting the transcription factor MtNLP1 controls both local and systemic response to N. 

Currently, peptides, miRNAs, and hormones are all known to mediate nitrogen demand signaling under low nitrate conditions by binding to the receptors or by affecting gene expression in root and or shoot [59,61,62] (see Figure 2b). A root generated peptide from the C-TERMINALLY ENCODED PEPTIDE (CEP) family, MtCEP1, is the only well-studied peptide signal of *M. truncatula* nitrogen demand signaling, exhibit negative effects on lateral root formation and positive effect on nodulation under low N conditions [61] (Figure 2b). CEP1 is generated in roots under low soil N availability and binds to the shoot receptor COMPACT ROOT (MtCRA2) to control nodulation from the shoot and lateral root formation from the root [63,64]. 

Recent research reveals some of the aspects of root competence for nodulation under low nitrogen conditions. The CEP1-CRA2 dependent enhanced expression of miR2111 in shoots (see AON section below) was shown to target the mRNAs of the genes *TOO MUCH LOVE 1* (*TML1*) and *TML2* in roots, lowering the transcript levels of both genes presumably to maintain susceptibility to symbiotic nodulation [59] (Figure 2b). A separate study showed that CEP1-activated MtCRA2 phosphorylates MtEIN2 (ETHYLENE INSENSITIVE 2), preventing its cleavage and repressing an ethylene response, thus promoting the root susceptibility to rhizobia [62] (Figure 2b). Exogenous ethylene treatment during the first 24 to 48 h of rhizobial inoculation is enough to suppress nodulation in wild type plants [65]. Although MtCRA2 is the common component in the systemic peptide-miRNA pathway and the root-localized ethylene pathway of maintaining root susceptibility to rhizobia, the exact mechanism of how the two pathways coordinate susceptibility is still unknown.

### 3.2. Autoregulation of Nodulation Signaling

In addition to local signaling, the existence of a long-distance systemic signaling mechanism controlling the nodule number was demonstrated by split-root experiments, in which prior inoculation of one-half of the root system suppressed nodulation in the other half [66]. Such long-distance signaling controlling overall nodule number depending on early nodulation events was termed “autoregulation of nodulation (AON)” [67]. AON is now known to involve root and shoot components and root-to-shoot-to-root signals through combined evidence from genetic and biochemical studies carried out by many researchers (Figure 2c). Split root experiments in *M. truncatula* demonstrated that AON occurs between two and three days after inoculation with rhizobia, and the same level of suppression is maintained for at least 15 days [50]. 

#### 3.2.1. Components of AON

AON involves receptors, modifying enzymes, and transcription factors that are known to act specifically from root or shoot to control nodule number. In addition to the local signaling, peptides, hormones, and miRNAs are signaling systemically. A defect in a component of AON reduces the suppression of nodulation, resulting in a hypernodulating phenotype [67]. Root to shoot reciprocal grafting experiments using hypernodulating mutants have provided insights into whether the action of an AON gene is root or shoot dependent and split root and grafting experiments (Y grafts) are used to determine the involvement of a root component in generating a root signal or receiving a shoot signal [50]. 

In *M. truncatula*, three hypernodulating mutants contain lesions in genes encoding components of the AON pathway: *sunn*, *rdn1*, and *crn.* The *sunn* mutants contain lesions in *SUPER NUMERARY NODULES (SUNN)* encoding a CLAVATA1-like leucine-rich receptor kinase [68,69], while *rdn1* mutants lack expression of *ROOT DETERMINED NODULATOR (MtRDN1)* encoding an arabinosyl transferase enzyme [58], and the *crn* mutant contains a *Tnt1* insertion in *CORYNE (MtCRN)* encoding a pseudokinase [70]. In *Lotus japonicus*, in addition to mutants in orthologs of *SUNN* and *MtRDN1* which result in hypernodulation, three cloned hypernodulators are described harboring defects in (i) *CLAVATA2 (LjCLV2)*, a receptor-like protein without a kinase domain [71], (ii) *KLAVIER (LjKLV)*, a receptor-like kinase [72,73] and (iii) *TOO MUCH LOVE (LjTML)*, a nuclear-localized Kelch repeat-containing F-box protein [74,75]. *M. truncatula* contains two sequence homologs of LjTML, MtTML1, and MtTML2; downregulation of either gene using RNAi resulted in a slight increase in nodule number [76]. The location of the effects generating the hypernodulation phenotype for all mutants except *rdn1* and *tml* are shoot-determined (reviewed in [46]). An *M. truncatula* shoot-determined hypernodulating mutant with an unknown causative mutation named *like sunn supernodulator* (*lss*) may function by epigenetic modification at the *SUNN* locus, as the defect is a lack of *SUNN* expression even though the *SUNN* sequence is a wild type [77]. The plasma-membrane-localized SUNN protein exists as a homomer, and in heteromeric form with MtCLV2 or MtCRN, hence it is likely to function in a receptor complex [70]. 

#### 3.2.2. Systemic Signals of AON

Two members of the CLE peptide family, MtCLE12 and MtCLE13 (LjCLE-RS1/LjCLE-RS2; GmRIC1/GmRIC2; PvRIC1/PvRIC2) are known root to shoot signals of AON (Figure 2c) [78,79,80]. In addition, MtCLE35 was recently identified as a root signal in response to rhizobia as well as high soil nitrate [55,56]. The mature CLEs functional in AON are post-transcriptionally modified 13 amino acid long peptides containing three arabinose residues added to the 7th hydroxyproline amino acid residue [81]. 

A mobile microRNA, miR2111, which is downregulated in *L. japonicus* shoots after rhizobial inoculation, moves from shoot to root through the phloem and functions as a shoot-to-root signal [82]. The expression of miR2111 in the shoot decreases after inoculation in *M. truncatula*, dependent on the shoot receptor kinase SUNN [59]. Rhizobial inoculation leads to reduced shoot to root auxin transport overlapping with the onset of AON in wild type plants but not in hypernodulating *sunn-1* plants, which have constitutively higher levels of the shoot to root auxin transport [83]. The study also found that the application of an auxin transport inhibitor at the shoot/root junction reduces nodule number in *sunn-1*, indicating that auxin can be a signal inhibiting nodulation. Rhizobial inoculation also results in upregulation of cytokinin biosynthesis in the shoot in a LjHAR1 dependent manner, which can inhibit nodulation; thus, cytokinin has also been proposed as a shoot-to-root signal [84]. Unlike the conserved and well-accepted CLE peptides as a signal for root-to-shoot signaling, agreement on a single shoot-to-root signal is lacking and the possibility of multiple shoot generated signals cannot be ruled out.

#### 3.2.3. Mechanism of AON

In response to rhizobia, expression of the transcription factor MtNIN is induced (see early symbiotic signaling and nodule organogenesis sections), which binds to the promoter of *MtCLE13* activating its transcription [36] (Figure 2c). In addition, CLE12 and CLE13 expression is also dependent on cytokinin through MtCRE1 [36,40,85]. The induction of CLE expression after rhizobial inoculation is the first known step of the AON pathway. Knockdown of *MtCLE12* and *MtCLE13* using RNAi results in a significant increase in nodule number [85]. Overexpression of *MtCLE12* or *MtCLE13* or *MtCLE35* in roots inhibits nodulation in wild type, depending on the SUNN receptor in the shoot, potentially through xylem mediated translocation of peptides [46,69,70,79]. The post-transcriptional modification of CLEs resulting in triarabinosylated peptides is essential for the function in the negative regulatory pathway to control nodules as demonstrated by the absence of a nodule suppression phenotype upon ectopic expression of non-arabinosylated CLEs [81]. The MtRDN1 mediated arabinosylation of MtCLE12, but not MtCLE13, is required for AON [58]. Similarly, MtCLE35 overexpression in *rdn1* mutant plants does not suppress nodulation, indicating a similar requirement of MtRDN1 mediated modification for its function in AON [56]. Whether or how MtCLE13 is modified is still unknown. There is likely a critical spatiotemporal regulation between the CLEs functioning in AON and those integrated with nitrogen demand signaling; further studies in the field are required to improve our current understanding.

The binding of a root-derived signal to a shoot receptor results in the generation of two potential shoot-derived signals mentioned above, cytokinin and miRNA2111 [59,82,84]. Rhizobial inoculation reduces the expression of miR2111 in the shoots (dependent on MtSUNN/LjHAR1), resulting in decreased abundance of the mature miR2111 in the roots, which in turn prevents degradation of *MtTML1* and *MtTML2* transcripts, leading to inhibition of further nodulation [59,82]. However, the mechanism of reduction in miR2111 expression by MtSUNN/LjHAR1 remains unknown. The enhanced cytokinin production in shoots after rhizobia inoculation [84] and the involvement of cytokinin through the cytokinin receptor MtCRE1 in the generation of CLE peptides [36,85] suggest a potential involvement of cytokinin in AON, but the transport of cytokinin to the roots and whether the cytokinin that affects the CLEs is shoot derived has not been established, due to technical difficulties in experimental design. Furthermore, while the altered shoot to root transport and root accumulation of auxin depending on MtSUNN is correlated with AON [54,83] (Figure 2c), the exact mechanism of how auxin controls the number of nodules in AON is unknown. 

Interestingly, hypernodulating mutants carrying mutations in *SUNN* and *RDN1* that lack early suppression of nodulation are also tolerant to environmental nitrate [77,86]. However, they do show a suppression effect at 10 days or more post-inoculation, an effect synchronous to the formation of mature nodules, indicating suppression by biologically fixed nitrogen occurs in these mutants [50], but again, the underlying mechanism is unknown. 

In addition to effects from auxin and cytokinin, high levels of ethylene suppress nodulation in wild type plants and in all hypernodulating mutants studied to date for this effect of ethylene in nodulation, with the exception of a mutant defective in ethylene signaling, *sickle* (*skl*) in *M. truncatula* [46,65]. The hypernodulation and lack of ethylene sensitivity to nodulation in *skl* are caused by a mutation in *M. truncatula* ortholog of Arabidopsis ethylene signaling protein EIN2 [87]. Ethylene sensitivity in *L. japonicus* requires two copies of *EIN2*, *LjEIN2-1*, and *LjEIN2-2*, for the conserved function [88]. The ethylene-mediated suppression of nodulation in most hypernodulating AON mutants contrasted with the *EIN* mutants suggests an independent role of ethylene and AON in controlling nodulation [65,68]. As opposed to maintaining susceptibility (described in nitrogen dependent control of nodulation, also see Figure 2b), rhizobial inoculation leads to rapid upregulation of ethylene biosynthesis that promotes MtEIN2 cleavage, activating the ethylene pathway to inhibit rhizobia infection [62]. Taken together, the data suggest that ethylene controls the number of nodules through MtEIN2 locally by controlling susceptibility to rhizobial infection independent of the AON pathway. 

## 4. Perspective

In summary, legumes balance the number of nodules formed with the plant’s need for N by the integration of the outputs of at least three signaling pathways. Both local and systemic, these pathways include a pathway for nodule initiation, an inhibitory pathway for nodule number (AON), and an inhibitory pathway based on N sufficiency. The more we know, the more questions there are to ask. For example, how do the specific cortical cells that become the nodule enter the cell cycle and divide into nodule primordia, while the adjacent cortical cell does not? Is this a point of regulation, and is it related to chromatin modification? AON occurs within 48 h of inoculation, but does it halt cell division that has already begun, or does it just prevent further initiation? N signaling has been linked to root growth in Arabidopsis through phosphorylation/dephosphorylation of PINs (auxin efflux carriers)-could this explain how N controls nodule formation as well? All of the hormones involved in regulation can be further regulated at the levels of synthesis, transport, and modification/degradation, suggesting areas ripe for future research. Finally, the observation that much of the regulation of nodule development may occur at the level of mRNA stability and translation [89] leaves much more to be discovered. As noted throughout this review, there are interesting hints about how cross-talk occurs between the three pathways, but precisely how the fine-tuning of nodule number is determined remains an open question. Research on the shared components between these pathways, the generation of mutants in multiple individual components of these pathways, and -omics experiments beyond transcriptomes hold the potential to fill in the picture of how nodule number is controlled.

## Figures and Tables

**Figure 1 ijms-22-01117-f001:**
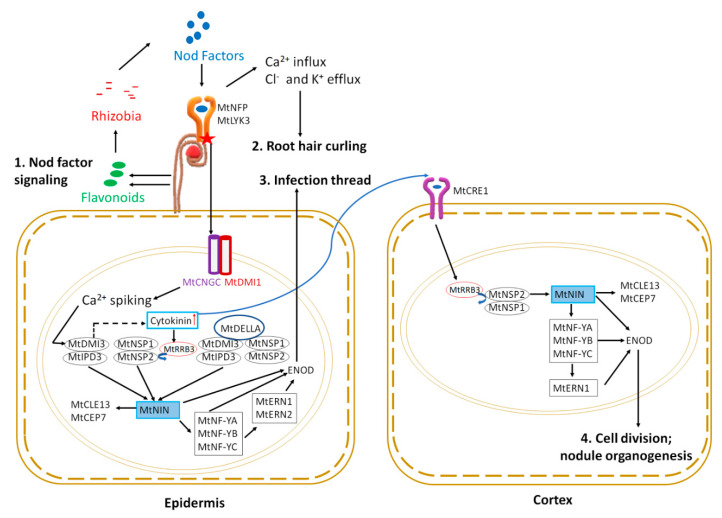
Early symbiotic signaling in epidermis and cortex during nodulation. Nod factor signaling: (1) the process starts with Nod factor signaling initiated by plants by producing flavonoids that attract rhizobia. Rhizobia produce Nod factors that bind to root hair receptors (MtNFP and MtLYK3), initiating the intracellular signaling cascade in the nucleus described in the text. Nuclear signaling leads to altered ion fluxes resulting in root hair curling (2). Simultaneously, Ca2+ spiking activates MtNIN through activation of a series of transcription factors; expression of early nodulin (ENOD) genes, such as *ENOD11* and *ENOD12*, facilitates infection thread progression (3). MtNIN also induces the expression of two small peptides, MtCEP7 and MtCLE13. Cytokinin signaling through MtCRE1 mediates MtNIN, MtNF-Y, and MtERN production in the cortex, which upregulates *ENOD* expression leading to cortical cell division; (4) and nodule organogenesis (see text for details).

**Figure 2 ijms-22-01117-f002:**
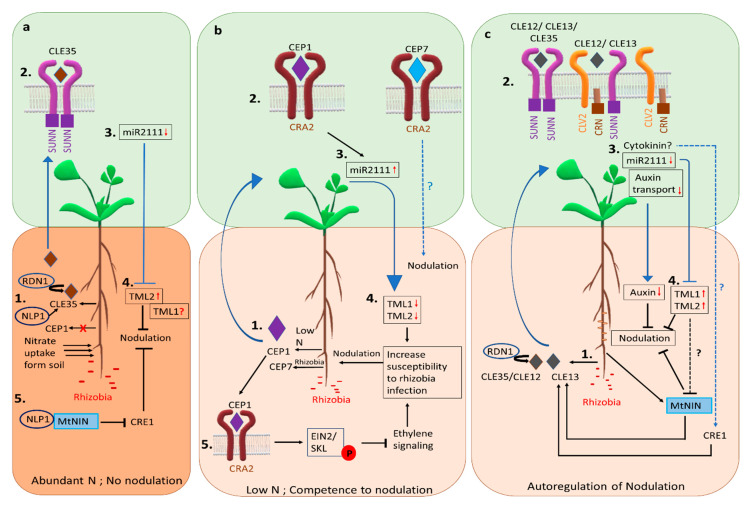
Mechanisms of controlling nodulation in *M. truncatula*. Under abundant soil N availability, plants take up nitrate from soil and do not form symbiotic nodules. (**a**) In response to nitrate (1), the expression of *MtCLE35* is induced in the roots, dependent on the transcription factor MtNLP1. The small peptide MtCLE35 reduces nodule number systemically dependent on MtSUNN (2). This results in reduced expression of miR2111 in the shoot (3), releasing the repression of the *MtTML2* transcript by miR2111 in the root (4). The reason for continued repression of *MtTML1*, which is also the target of miR2111, is not understood; whether MtTML1 is involved in nitrate control of nodulation is not clear (see text). Locally in the roots (5), MtNLP1 binds to MtNIN, inhibiting MtCRE1 expression to inhibit nodulation. (**b**) Under N limited conditions, the small peptide MtCEP1 is generated in the roots (1), which binds to the MtCRA2 receptor in the shoot (2), upregulating the expression of miR2111 in the shoot (3). Evidence suggests miR2111 is transported through the phloem to the roots, increasing the abundance of mature miR2111 in the roots. Mature miR2111 targets and lowers the transcript levels of *MtTML1* and *MtTML2* (4), increasing susceptibility to rhizobia and reducing AON. MtCEP1 activates MtCRA2 in the roots to phosphorylate MtEIN2, preventing its cleavage, thus repressing the ethylene response and promoting susceptibility to rhizobia (5). Another peptide, MtCEP7, generated in response to rhizobia, promotes nodulation dependent on the MtCRA2 receptor. (**c**) autoregulation of nodulation (AON). Rhizobial inoculation leading to nodule initiation results in the generation of MtCLE12 and MtCLE13 signaling peptides in the roots (1), which are transported in the xylem and bind to a shoot receptor complex containing MtSUNN, MtCRA2, and MtCRN (2). Together, the complex causes downregulation of miRNA2111 expression in the shoot (3). The result, perhaps through the transport of cytokinin as well as miRNA2111, is decreased miR2111 abundance in the roots and increased transcript levels of its targets *MtTML1* and *MtTML2* (4), inhibiting further nodulation. Another small peptide MtCLE35, which is also induced by high nitrate, controls the nodule number depending on MtRDN1 and MtSUNN in a similar manner as MtCLE12. MtNIN, involved in nodule organogenesis, also activates *MtCLE13* expression to initiate AON. MtNIN might be under MtTML1 and MtTML2 regulation, maintaining a feedback control between nodule organogenesis and AON. Another potential shoot to root signal, cytokinin, may function through MtCRE1, which is required for *MtCLE13* expression. Dashed lines represent proposed mechanisms; blue lines indicate systemic action and black lines indicate local events. All gene names in the figure are *M. truncatula* gene names, shown without the initial *Mt* for simplicity.

## Data Availability

This review did not report any new data.

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
