# Peer review of "The Regulation of Nodule Number in Legumes Is a Balance of Three Signal Transduction Pathways"

_ijms, 2021, doi:10.3390/ijms22031117_

Round 1
Reviewer 1 Report
Plant growth-promoting rhizobacteria (PGPR) has been reported that enhances plant growth and development. Nitrogen-fixing soil microbes, especially in arbuscular mycorrhizal fungi (AMF) and nitrogen-fixing bacteria, are able to form a symbiotic relationship with leguminous plants. The interaction between rhizobia and leguminous plants is to form nodules on the plant root, within which the rhizobia convert atmospheric nitrogen into ammonia that can be used by the plant. In the manuscript, the authors present a pretty good background of Genetic and Molecular Mechanisms in Legume-Rhizobium Interactions, as well as a summary of the recent studies.
To sum up, the review manuscript was well organized and well written. In addition, the authors present a very interesting topic. There are currently many interests in legume-rhizobium symbiotic interactions. It would be of wide interest to the crop community, the legumes industry, and the International Journal of Molecular Sciences (IJMS) readers. However, I have some concerns about the manuscript, before publication:
Major concerns:
1. In line 94, the author described that “three members of MtCNGC (CYCLIC NUCLEOTIDE GATED CHANNEL; also called MtDMI3): CNGC15a, CNGC15b, and CNGC15c form a complex with the potassium gated ion channel MtDMI1”. However, DMI3 has been documented as calcium/CaM-dependent protein kinase (CCaMK not CNGC15s) which functions downstream of nuclear calcium spiking and is required for Nod factor signal transduction, nodulation, as well as mycorrhization (doi.org/10.1038/nature22009 and doi.org/10.1016/j.pbi.2017.06.003). Please revised the sentence.
2. The authors mentioned that a balance of three signal transduction pathways in the title: Early symbiotic signaling; Nitrogen in soil suppress nodule formation; autoregulation of nodulation. However, the authors did not clearly describe the three signal transduction pathways. It is better to highlight the three signal transduction pathways in the Abstract and/or Introduction section.
3. The manuscript was well presented; however, the perspective section was missed. The authors would offer some views for future studies and IJMS readers.
Minor concerns
1. The authors provided an interesting Figure 1. However, the legion of figure 1 missed the detail. More information would be very helpful for readers to understand the figure, such as 1. Nod factor signaling; 2. Root hair curling; 3. Infection thread; 4. Cell division and nodule organogenesis.
2. I suggested to remove the subtitle “2 review” and directly use “1 Initiation and formation of symbiotic nodules”. Moreover, please revise 5 conclusions” accordingly.
3. “Line 175, ENOD11 promoter” ENID11 is gene name, hence it should be italic
4. Line 194 symbiotic nitrogen fixation (SNF) was defined before line 43.
5. Although most abbreviations were present very well in the manuscript, few still need to be taken care of, such as line 92 MtLYK3. Please check all of the abbreviations in the manuscript.
Author Response
Reviewer 1
Thank you for your helpful suggestions!
Major concerns
- MtDMI3 was a typo, the sentence also got garbled somehow. We changed it reflect the three calcium channels that interact with the potassium channel DMI1.
- We did name the three pathways in the abstract (line12-14) and Introduction (lines 50- about 57). However, they were not clearly stated in form we use in the rest of the review and lines 50-57 and the abstract have been revised to reflect the reviewers request.
- The conclusions section has been retitled “perspectives” and expanded with unanswered questions.
Minor concerns
- We have edited the figure legend to align with the numbered processes within the figure and added some clarification at the request of reviewer 3.
- This was an artifact of using the journal template-we have removed the heading and taken the reviewer’s suggestion. We also revised “conclusions” as mentioned above in #3.
- Corrected typo on ENOD11 (italics)
- Removed ‘symbiotic nitrogen fixation” and replaced all mentions after first with SNF-added SNF to abbreviations list
- Checked all abbreviations- we corrected LYK; we are not certain if we missed any more. We also made sure N was used consistently.
Reviewer 2 Report
This review is timely and of interest for the plant science community. The paper is written in clear and concise way and illustrations are well done.
I only have a few suggesstions:
1) Lines 28/29: Indeed NO3- and NH4+ are the main N forms taken up by plants. However plants are able to take up others like NO2- or urea. Therefore, please write that NO3- and NH4+ are the main forms taken up by plants.
2) Line 42. Briefly comment here that currently only a small proportion of commercial legume crop production relies on biological N fixation.
3) Autoregulation and Nodulation should be written in lower case
4) Section 2.1. a scheme illustrating the decription of the developmental zones woud be helpfu here.
5) Line 84: What is known about the release? It is constitutive or induced? Does it respond to N scarcity and is it suppressed by external NO3- or NH4+?
Author Response
Thank you for your helpful suggestions & questions
Reviewer 2
- Added “main forms” as suggested to line 28/29
- We have added the reviewers observation to line 42.
- We have corrected all capitalization when found.
- We cited a review that has a figure showing the nodule zones in the initial manuscript. This is such a common diagram that it even appears on Wikipedia, and we are not sure we could produce another version of it that didn’t infringe on a copyright. Since no other reviewer felt this would help, if the reviewer feels strongly about it, we could link to the Wiki diagram: https://en.wikipedia.org/wiki/Root_nodule.
- We thank the reviewer for raising this question. There is no evidence, based on a 2020 comprehensive review of flavonoids in symbiosis in this journal, that the release of flavonoids into the soil is anything but constitutive-we have edited line 84 to reference this review and added it to the reference list.
Reviewer 3 Report
Very comprehensive review, but some corrections are required.
Below there are some points:
Abstracts: it will be nice to mention these three transduction pathways more clear on line 12. Introduction: I would suggest starting with the role of nitrogen as a building material for the plant body, not with NO. Despite NO is very important in defence response, nitrogen is more important in nutrition. Line 32: Tg? Lines 37-39: the sentence is not clear, this part does not fit well. What is „contains 55-70% of fixed nitrogen in its aboveground parts“? Line 48: why these three species? Line 58: why subtitle review? Please, choose the better one. Line 74: it is better to move this sentence to line 69: „L. japonicus is studied as a determina-te legume model“. Figure 1: there are some exact repetitions between figure 1 legends and the main text. From the epidermis panel, it is not clear what do you mean as cytokinin: synthesis, sig-naling, or else. From the cortex panel, it is not clear what do you mean as cell division. Is it mitosis or chromatin remodeling that made cell competent to the cell cycle? Lines 158-160: the sentence is not clear. For example, what do you mean as : „simultaneous with cell division leading to development of nodule“: site of cell cycle activation in cortex or cell cycle activity in each nodule? Lines 160-168: please, be more precise about the species you are mentioned: LjLHK1 is from L.japonicus, but later, you mention Medicago. There are two different nodules with different cortex cell activation (which have different epigenetic statuses in the root). Line 201: Please, clarify which nitrogen do you mean? It was shown recently that nitrogen has an effect on auxin sig-naling/distribution. https://www.embopress.org/doi/full/10.15252/embj.2020106862 Moreover, it is known that the nitrogen level has a significant effect of root cell epigenetic status. It will be nice to link it with the nodulation process or point out this possibility as the mechanism. Line222: please, clarify which root cell do you mean: epidermis, cortex 1, cortex 2, cortex 3 or so… Figure 2, panel C: what do you mean as cytokinin: hormone biosynthesis, signaling or else? Line 349: Cytokinin is not a signal, it is a hormone. Please, be more precise. MiR2111 can not be compare with hormone. Some general point: there are only a few citations from recent years; the majority are five years old or more. In the review, it is better to cite the most recent advanced publications, while some older ones can also be present, but not more than 20-25%. There are two recent publications, for example, that did not mention but give more under-standing of the topic. Azarakhsh, M., Rumyantsev, A. M., Lebedeva, M. A., & Lutova, L. A. (2020). Cytokinin biosynthesis genes expressed during nodule organogenesis are directly regulated by the KNOX3 protein in Medicago truncatula. PloS one, 15(4), e0232352. Lebedeva, M., Azarakhsh, M., Yashenkova, Y., & Lutova, L. (2020). Nitrate-Induced CLE Peptide Systemically Inhibits Nodulation in Medicago truncatula. Plants, 9(11), 1456.
Please, improve layout: space between lines, citation number after sentence, but not before next etc.
Author Response
Thank you for your attention to detail on this review.
Since you did not number suggestions or indicate major versus minor concerns, we are responding to the almost 20 comments made in order of mention. At times we were unable to determine what the specific change was that was being requested, and if we have not addressed an issue, it is because we need more clarity. However, your questions prompted us to add more text and additional references based on unresolved questions brought up by the reviewer, and include a manuscript that was published the day the reviews were returned to us.
- Abstract not listing pathways: Reviewer 1 mentioned this as well and we have edited the abstract and introduction to mention the three pathways in a consistent manner.
- Request to start with nitrogen as a major constituent of the plant and then mention NO as a signaling molecule-we are confused by the comment, as that is exactly the way the first sentence is written. No changes were made; we require further clarification if this is important.
- Line 32 “Tg?” Tg is the scientific notation for Teragrams, and like mg and ng, MDPI approves it as needing no explanation.
- Line37-39. Since the reviewer was unable to determine that soybean was the subject of the second half of this compound sentence, we split the sentence into two sentences and changed the phrasing to make it clear that soybean has more nitrogen in its aboveground parts than non-legumes.
- Line 48-“why these three species?” We actually chose four species as examples that span the globe without being exhaustive. Since we add the phrase “among the many” we don’t believe an explanation of why these on the list is required, especially since none of them are dealt with in detail in the review. If the reviewer would like more or fewer examples of crops we could add them, just clarify which crops if any should be the list.
- Line 58. This was an artifact of using the journal template and all three reviewers pointed it out-it has been removed.
- Moving sentence on line 74 to line 69: We agree the sentence as placed is awkward, but moving it to the location requested breaks up the flow of the description of an indeterminant nodule. Instead we removed the sentence in question and edited the next line to indicate there are models for each system and our focus is on truncatula.
- “Figure 1:there are some exact repetitions between Figure 1 and the main text.” Since Reviewer 1 asked for more detail and division of the legend, and we assume Reviewer 3 is talking about the legend, we have edited the legend to address both reviewers comments.
- The comment about “cytokinin not being clear -synthesis, signaling or else?” is also confusing to us. It is known the hormone cytokinin is involved in the process. That is all that is meant. It is likely synthesis, movement, and reception are part of how cytokinin affects nodulation but since this is a black box at the moment, we do not indicate more than “cytokinin”. However, we thank the reviewer for pointing this out as we have added this unknown to the perspectives section.
- We are also confused by the request to indicate “whether it is mitosis or chromatin remodeling that make the cell competent to the cell cycle?” By definition, a cell in mitosis is dividing, thus in the M phase of the cell cycle. Does the reviewer mean leaving G0 as “competent to the cell cycle”? We added a recent report of truncatula histone deacetylases being critical for nodulation but it is not peer reviewed. Whether the small number of cells that undergo mitosis at nodule initiation in M. truncatula undergo chromatin remodeling to initiate the move from G0 to G1 stage is currently unknown and technically difficult to determine. However this very interesting paper on deactylases in M truncatula by Ton Bisseling on BioRV (thus preliminary) indicates chromatin remodeling is important to nodule initiation through regulation of hormone pathways. We have added a paragraph covering this in lines 164-173.
- Lines 158-160- Is “simultaneous with cell division leading to the development of the nodule: site of cell cycle activation in the cortex or cell cycle activity in each nodule?” Since the cell cycle activation in the cortex IS by definition the cell cycle activity in the nodule (this is the tissue nodules are formed from) we did not make the distinction, and the short answer is “we don’t know yet”. We revised the sentence before cell division to make the lack of distinction a little clearer. Again, thank you for pointing out an interesting distinction, which we added to the perspectives
- A request to be clear about species. We separated the sentences about LjLHK1 and MtCRE1, since they are the same molecule, but one occurs in indeterminate nodules and one in indeterminant nodules and they may function differently. As noted at the beginning of the review we are focused on truncatula and only note other species where relevant (see modification in comment #7)-we mention LjHDK here because it led to the discovery of MtCRE1 which proved pivotal.
- Line 201 “which nitrogen?” We have changed the heading to “Control of nodulation based on nitrogen need and soil availability” since we discuss both. We are thankful for the reviewer’s provision of a reference for a paper, published since our manuscript submission, linking auxin transport and nitrogen in plant roots and we added this new information to the review (line 195).
- Line 222 “Please clarify which root cell do you mean?” We used the generic term “root cells” because the source of the experiment that shows this, (Lin, et al. 2018) used it. From the pictures in the Lin manuscript, it is clear that at least the outer cortical cells have this localization, but it is impossible to see more than a few cell layers into the root so we stayed with the generic “root cells”.
- “Figure 2 panel C what do you mean as Cytokinin-hormone biosynthesis, signaling, or something else?” See comment #8. We realized we did not specifically mention cytokinin’s suggested role in the figure legend, and only in the text, so we added “perhaps through transport of cytokinin as well as miRNA2111”, to the figure legend.
- Line 349 “Cytokinin is not a signal, it’s a hormone.” We beg to differ. Hormones are signals, as are miRNAs, which the reviewer also decries as a signal. The definition of a molecular signal as defined by the Journal of Cell Signaling is a molecule which transfers information between cells, usually by binding to another molecule. (https://www.longdom.org/scholarly/molecular-signaling-journals-articles-ppts-list-3204.html). Both of these molecules are signals and there is evidence suggesting both can be transported systemically.
- General point-the reviewer suggests the percentage of citations since 2015 is low and suggests two papers by the same group to add, one of which we already cited (#48 in original manuscript). We were pleased to add the uncited cytokinin work suggested, as well as a third paper by this group into the discussion of cytokinin (Lines 140-145 in revised manuscript), as an indication of multiple ways cytokinin is involved in nodulation. Unrelated, we also added a VERY recent 2021 paper on N signaling in nodulation and revised the figure, because the findings change our understanding of this process. Since a review cites the initial discoveries in a field, a good review will cite foundational papers in a field that change our understanding, as well as new ones, and our citations were driven by the important discoveries, not a desire to mention every recent paper no matter how incremental. We checked, and with the addition of 8 references added in addressing questions posed by reviewers, over a third of our citations are less than 5 years old and two are less than 2 weeks old. Unless there are suggestions as to what we have missed that is specifically relevant to nodule number signaling, we see no way to “fix” this.
- Improve layout: We have used the template and spacing the journal provides. While some have strong preferences about the position of a reference within a sentence, when multiple statements within a complex sentence are supported by different citations, mid-sentence citation is the only way to help the reader link the citation to the statement, and it was done for clarity. MDPI has no statement about this in its guide to references (version 5) and the other reviewers did not mention this, so we defer to the editor for placement of references within a sentence versus all in a group at the end.
Round 2
Reviewer 3 Report
Thank you, all points were answered properly. The paper can be accepted in the present form.
You are right; in early events of nodulation, change in chromatin accessibility is a key step that allows nodule induction (G0-G1 transition). In situ 3D assay give you information.